# Eye Gaze Markers Indicate Visual Attention to Threatening Images in Individuals with Chronic Back Pain

**DOI:** 10.3390/jcm8010031

**Published:** 2018-12-31

**Authors:** Zoë C. Franklin, Paul S. Holmes, Neil E. Fowler

**Affiliations:** 1Musculoskeletal Science and Sports Medicine Research Centre, Manchester Metropolitan University, Manchester M15 6BH, UK; p.s.holmes@mmu.ac.uk; 2Vice Chancellor’s Office, University of Salford, Salford M5 4WT, UK; n.fowler1@salford.ac.uk

**Keywords:** chronic pain, attentional biases, eye gaze

## Abstract

Research into attentional biases and threatening, pain-related information has primarily been investigated using reaction time as the dependent variable. This study aimed to extend previous research to provide a more in depth investigation of chronic back pain and individuals’ attention to emotional stimuli by recording eye movement behavior. Individuals with chronic back pain (*n* = 18) were recruited from a back rehabilitation program and age and sex matched against 17 non-symptomatic controls. Participants’ eye movements were recorded whilst they completed a dot probe task, which included back pain specific threatening images and neutral images. There were no significant differences between chronic pain and control participants in attentional biases recorded using reaction time from the dot probe task. Chronic pain participants, however, demonstrated a significantly higher percentage of fixations, larger pupil diameter, a longer average fixation duration and faster first fixation to threatening compared to neutral images. They also had a significantly longer average fixation duration and larger pupil diameter to threatening images compared to control participants. The findings of this study suggest eye gaze metrics may provide a more sensitive measure of attentional biases in chronic pain populations. These findings may have important therapeutic implications for the patient and therapist.

## 1. Introduction

Attentional biases are a selective attention towards or away from a stimulus, which is both specific and salient to an individual’s current environment and situation [1] and can result in a variety of cognitive, behavioral and physiological responses. Excessive attentional biases towards pain have been hypothesized to contribute towards the promotion of pain-related anxiety, fear of pain-related activity, physical disability and exacerbations in the pain experience [2]. Attentional biases towards pain-related stimuli have been proposed in theories of attention and pain [3,4]. The pain-specific models explain an individual’s attentional response to the presence of pain-related stimuli. Todd et al. [5] proposed the threat interpretation model of attentional biases, suggesting that there is a relationship between an individual’s interpretation of threat, which then influences their attentional bias towards it. As threat interpretation increases, initial vigilance towards pain-related stimuli increases; the level of the threat then influences whether the individual is able to disengage from the threat or avoid the threat.

Meta-analyses [6,7,8] have reported that pain participants have an attentional bias towards threat related information compared to controls, although with low effect sizes of 0.1–0.3 which may be due, in part, to the lack of consistency with the type of pain-related stimuli (e.g., words or pictures; sensory or affective pain). Within the most recent review [8], the majority of studies used word-based stimuli or pain-related faces, rather than pictures associated with movement. Kourtzi and Kanwasher [9] have identified that pictures which have implied physical movement lead to greater activation of the extended motor system, compared to images without implied movement. Therefore, it could be suggested that the increased motor activity associated with movement-related images might represent a marker of a more valid cognitive response to movement-related emotional stimuli.

Attentional biases have been assessed, traditionally, using the dot probe paradigm [10]. This technique captures a momentary cross-section of attention at the stimulus offset. Therefore, the method does not indicate duration of attention or attentional exertion to stimuli [11]. The use of eye-tracking equipment, however, can address some of these limitations and allows researchers to record location and duration of gaze fixations. In addition, eye-tracking also provides the opportunity to measure pupil diameter as an index of attentional effort [12]. Eye gaze markers can, therefore, provide researchers with a range of additional metrics to improve the understanding of attentional biases in chronic pain patients.

Several studies have utilised the dot probe method in conjunction with eye tracking and identified a bias towards pain-related information. Unfortunately, however, these studies were all conducted with pain-free undergraduate students, using pain-related words [13,14] or faces [15,16,17]. While these studies do provide some insight into the mechanisms associated with attention to pain-related information, further research is needed to be conducted in patients suffering from chronic pain. To our knowledge, only three studies have used eye movements to examine attentional biases to pictorial stimuli, and these were with chronic headache populations [18,19] and mixed chronic pain groups [20]. Both Liossi et al. [18] and Schoth et al. [19] used a visual search task to assess the attentional biases of chronic headache patients. Participants were shown four facial expressions (pain, angry, happy and neutral). Both studies [18,19] identified that the pain group demonstrated a significantly higher proportion of initial fixations on the pain face, compared to the other facial expressions. Furthermore, Liossi et al. [18] found that patients had an initial shift in their attention towards pain stimuli, but then maintained their gaze on happy” images. There was no evidence that pain patients maintained their gaze on pain related images. These findings support the theoretical models of pain and attentional biases [3,4,5] and propose that attentional biases may play an important role in the maintenance of chronic headache. Fashler and Katz [20] investigated the attentional biases of undergraduate students who were experiencing a variety of chronic pain conditions (e.g., neck/back, migraine, ankle/knee, stomach, hip, arm, eye and jaw pain). Participants completed a dot probe task using injury related (e.g., needle being inserted into the skin, black eye, open wound, burned skin) and neutral images while their eye movements were recorded. Reaction time results revealed that chronic pain individuals responded faster to neutral stimuli in contrast to the injury related images. In contrast, the eye tracking data demonstrated that chronic pain individuals maintained attention towards injury related pictures. Supporting Todd et al.’s [5] theory that as the interpretation of threat increases, vigilance towards the threat also increases. To date, however, no study has investigated the attentional biases of a chronic back pain patient population using both the dot probe paradigm and eye tracking approaches.

The aim of this study, therefore, was to provide a more in depth investigation of chronic back pain patients’ attention to pain-related images in comparison to non-symptomatic controls by recording eye movement behavior whilst participants also completed a dot probe task. We hypothesized that chronic back pain participants would have: (i) a significantly higher percentage of fixations to threatening stimuli compared to controls; (ii) a longer average fixation duration to threatening images; and (iii) exhibit a faster reaction time to threatening images in the dot probe task.

## 2. Method

### 2.1. Participants

Participants were recruited from a back rehabilitation program at a UK NHS trust (*n* = 18) and an age and sex matched non-symptomatic control group was recruited from the university and local area (*n* = 17). Chronic pain participants had been suffering from back pain for a minimum of three months. Non-symptomatic controls were recruited through advertisements through unsolicited noticeboards and electronic advertising. Ethical approval was granted by NRES Committee North West Greater Manchester Central and by Manchester Metropolitan University Ethics Committee. All participants provided written informed consent to take part in this study. Inclusion criteria for the back pain group were: (i) over 18 years of age; (ii) a referral to a hospital-based back pain management program for non-specific musculoskeletal pain; (iii) pain duration of >3 months; and (iv) normal or corrected to normal vision. Inclusion criteria for the control group were: (i) over 18 years of age; and (ii) normal or corrected to normal vision. Exclusion criteria for the control group were: (i) any form of current or recent chronic or recurrent pain; and (ii) regular (daily or near daily) use of any form of analgesic medication.

### 2.2. Materials

#### 2.2.1. Dot Probe Paradigm

All participants completed a dot probe task comprising 20 practice trials and 150 experimental trials (100 threat-neutral, 50 neutral-neutral). The threat images were taken from the Photograph Series of Daily Activities (PHODA) image bank [21] (back pain specific and showing movements known to be associated as threatening and evoking pain or pain-related fear, for example lifting or bending tasks). The neutral images were taken from the International Affective Picture System (IAPS) [22] and included images of neutral activities, faces and inanimate objects. The presentation of images were randomized for each participant. Dot probe stimuli were presented on a 23-inch screen (HP EliteDisplay E231, Hewlett-Packard Company, Palo Alto, CA, USA) with a 1920 × 1080-pixel resolution and a 100 Hz refresh rate. Participants were told to engage actively with the pictures that were presented to them on the screen. Each trial began with a central fixation cross presented for 500 ms, followed by an image pair on the left and right side of the screen, either threat-neutral or neutral-neutral pairs presented for 500 ms. Following presentation of the image pair, a probe stimulus (a pair of dots either vertical or horizontal) was presented in the location of either the emotional or the neutral image and remained displayed until the patient/participant responded (see Figure 1). Participants were instructed to press, as quickly and accurately as possible, one of two keys to identify the probe presented. The inter-trial interval varied randomly between 500 and 1250 ms. Response times shorter than 200 ms or longer than 1200 ms were removed from the data. Incorrect responses were also excluded from the analysis. Errors and outliers accounted for 2.5% of the data.

Congruent (e.g., the target followed the emotional picture) and incongruent (e.g., the target followed the neutral picture) attentional bias scores for threatening images relative to neutral were calculated for each participant from the response time data using the formula:Congruent = ((Trpr + Tlpl)/2) − ((Nrpr + Nlpr + Nrpl + Nlpl)/4)(1)
Incongruent = ((Tlpr + Trpl)/2) − ((Nrpr + Nlpr + Nrpl + Nlpl)/4)(2)

T = threat, N = neutral, p = probe, r = right position, l = left position.

#### 2.2.2. Eye-Gaze

While participants completed the dot probe paradigm, their eye movements were also recorded. Eye movement data were recorded with an Applied Science Laboratories Mobile Eye System (ASL; Bedford, MA, USA) using a dark pupil tracking technique throughout the dot probe paradigm. This method uses the relationship between the pupil and a reflection from the cornea to calculate the point of gaze in relation to an external scene camera. The ASL software computes the relationship between the pupil and cornea to locate gaze within a scene at a sampling rate of 30 Hz. The system has an accuracy of 0.5° of visual angle, a resolution of 0.10° of visual angle, and a visual range of 50° horizontal and 40° vertical.

Previous eye gaze research has assessed: (i) percentage of fixations on the threat or neutral stimuli; (ii) average pupil diameter when fixating on the threat and neutral stimuli; (iii) average time spent fixating on the threat or neutral stimuli; and (iv) first fixation time on either the threat or the neutral stimuli. In this study, visual fixations were defined as maintaining gaze on a specific location on the screen for at least 100ms and a maximum fixation radius of 1°, as employed in previous studies [13,16].

### 2.3. Procedures

Participants were asked to sit at a desk in a black booth and facing the screen approximately 60 cm in front of them and at eye level. A desk mounted chin rest was used to reduce participants’ head movements ensuring that participants’ eyes were level with the middle of the monitor on which the stimuli were presented. This ensured that each participant’s eyes were in the same location relative to the camera and monitor. A 9-point calibration check was used prior to the start of testing. A drift check was conducted before each trial and recalibration performed when necessary. Participants were instructed to look at the fixation cross before each trial to standardize the starting location of their eye gaze and were told to engage actively with the pictures that were presented to them on the screen. Participants provided demographic information after testing was completed to allow for age and sex matching.

### 2.4. Preparation of Eye Gaze Data

Eye gaze data were analyzed using ASL Results Plus (Applied Science Laboratories, Bedford, MA, USA). Each trial was parsed into 150 (100 threat-neutral, 50 neutral-neutral) separate trials. Individual trials were then analyzed by drawing two separate areas of interest (AOIs) around the threatening and neutral images. From this, the number of fixations, average fixation duration, and pupil diameter when fixating in each AOI were calculated. A fixation was defined as any gaze that remained stable (within 1 degree of visual angle) for a duration of over 100 ms. In accordance with previous studies [23], participants with missing data of more than 15% over the 150 trials were excluded from the study. Based on this criterion, no participants were excluded from the study. Due to technical difficulties with the eye tracking equipment, no eye movement data were recorded for one of the non-symptomatic control participants and this participant was excluded from further analysis.

### 2.5. Data Analysis

A series of 2 × 2 analysis of variances (ANOVA) were conducted on the eye gaze data (percentage fixation, average pupil diameter, average fixation duration, first fixation time), with group (patient, control) as a between participants independent variable, and image type (threat or neutral image) as a within participants variable. A 2 × 2 ANOVA of the probe response time data with group (patient, controls) as the between variable, and probe position (probe in same versus different location to threatening image) as the within participants variable. Bonferroni *post hoc* analyses were used where needed to clarify significant main effects and interactions. Alpha was set at *p* < 0.05 and effect sizes were calculated using Cohen’s *d*.

## 3. Results

### 3.1. Group Characteristics

The chronic pain and control groups did not differ significantly in sex ratio (chronic pain: 12 (66%) female, control group 11 (64%) female, *χ^2^* = 0.15, *p*= 0.90) or age (chronic pain group: M = 46.72, SD ± 9.97 years; control group: M = 40.47, SD ± 9.23 years, *t*(33) = −1.92, *p* = 0.07). 

### 3.2. Power Analysis

Post-hoc power analyses [24] were conducted for the dot probe and eye movement data. The power analysis results for the dot probe data were: power = 0.64 (α = 0.05; β = 0.36) *d* = 0.6, for number of fixations to threat; power = 0.82 (α = 0.05; β = 0.18) *d* = 0.8, and neutral images; power = 0.83 (α = 0.05; β = 0.17) *d* = 0.9. The power analysis results for the average pupil diameter; power = 0.76 (α = 0.05; β = 0.24) *d* = 0.8; average fixation duration; power = 0.86 (α = 0.05; β = 0.14) *d* = 0.9; and for total fixation duration; power = 0.99 (α = 0.05; β = 0.01) *d* = 1.6. Therefore, despite the relatively modest sample size there was sufficient power to have confidence in the findings from the study.

### 3.3. Eye Gaze Reliability Analysis

Internal consistency, reflecting the interrelatedness of items on a test was calculated using Cronbach’s alpha for both the control group and the patient group for each outcome variable. For both patient and control groups there was high reliability for percentage fixation count, pupil diameter and first fixation time (Cronbach’s α = 0.73–0.93). Average fixation duration subscale, however, had a relatively low reliability, Cronbach’s α = 0.62–0.64 (see Table 1).

### 3.4. Percentage Fixation Count

The results of the two way mixed ANOVA showed a significant interaction between participant group and image type, (*F* (1, 33) = 32.01, *p* < 0.001) and a significant main effect of stimuli type, (*F* (1, 33) = 11.05, *p* = 0.002). Pairwise comparisons indicated that chronic pain individuals attended to threat images (M = 34.07, SD ± 10.58) significantly more than neutral stimuli (M = 18.27, SD ± 9.32); *t*(17) = 6.25, *p* < 0.001, *d* = 1.57, 95% CI [0.80, 2.29] (see Figure 2). For the control group, there was no significant difference between the percentage fixation count to threat (M = 21.61, SD = 5.97) or neutral (M = 25.72, SD ± 8.39) stimuli; *t*(16) = −1.69, *p* = 0.11, *d=* 0.5, 95% CI [−1.24, 0.13].

### 3.5. Average Pupil Diameter

A significant main effect of group was found (*F* (1,33) = 23.71, *p* = 0.0001); *t*-tests revealed that individuals in the pain group (M = 5.69, SD ± 0.18 mm) had a significantly larger pupil diameter compared to controls (M = 4.62, SD ± 0.18 mm), *d =* 0.5, 95% CI [4.29, 7.32]. A significant main effect for stimuli type was found (*F* (1, 33) = 11.65, *p* = 0.002); *t*-tests showed that participants had a significantly larger pupil diameter when attending to threatening (M = 5.30, SD ± 1.33 mm) compared to neutral (M = 4.64, SD ± 1.23mm) stimuli, *t*(34) = 2.71, *p* = 0.01, *d*= 0.9, 95% CI [−0.16, 1.17] (Figure 3). There was no significant interaction effect found (*F* (1,33) = 0.10, *p* = 0.749).

### 3.6. Average Fixation Duration

A significant group x image type interaction *F* (1, 33) = 10.90, *p* = 0.02, and a significant main effect of group *F* (1, 33) = 4.72, *p* = 0.03 was found. T-tests revealed that pain participants demonstrated a significantly higher average fixation duration on threatening (M = 219.18, SD ± 53.06 ms) compared to neutral stimuli (M = 185.04, SD ± 38.68 ms); *t*(34) = −2.16, *p* = 0.03, *d =* 0.8, 95% CI [−0.06, −0.001] (Figure 4). There was no significant difference between the control group’s average fixation duration towards threatening or neutral stimuli. T-tests revealed that the pain group (M = 219.18, SD ± 53.06 ms) had a significantly longer average fixation duration to threatening stimuli compared to controls (M = 174.88, SD ± 17.36 ms); *t*(68)= −3.35, *p* = 0.03, *d =* 0.9, 95% CI [−0.04, −0.004]. 

### 3.7. First Fixation Time

There was a significant interaction of group x first fixation time (*F* (1,33) = 35.21, *p* = 0.0001), and a significant main effect of group (*F* (1,33) = 27.00, *p* = 0.0001. Pairwise comparisons indicated that patients made significantly faster first fixations to threatening images, (*t*(33) = 8.90, *p* = 0.0001, *d* = 0.7, 95% CI [−3.59, −1.76] compared to controls (patient M = 124.98 ms, SD = 65.67; control M = 210.24, SD = 28.21) (Figure 5). The control group made significantly faster first fixations to the neutral stimuli compared to the threatening, (*t*(16) = 3.97, *p* = 0.001, *d* = 1.02, 95% CI [0.28, 1.71]). In contrast, the pain patients made significantly faster first fixations to the threatening image type compared to the neutral image type (*t*(17) = 4.44, *p* = 0.0001, *d* = 0.9, 95% CI [−1.47, −0.27]). There was no significant main effect of image type (*F* (1,33) = 5.36, *p* = 0.469). 

### 3.8. Dot Probe Response Time Measures

There was no significant interaction of group x attentional bias *(F* (1,33) = 3.17, *p* = 0.08), and no significant main effect of group (*F* (1, 33) = 1.69, *p* = 0.20) or probe position (*F* (1, 33) = 4.60, *p* = 0.12). There was no significant difference between attentional bias to threatening images for either congruent (*t*(33) = 1.79, *p* = 0.08, *d* = 0.6, 95% CI [−3.35, 51.14]) or incongruent (*t*(33) = 0.47, *p* = 0.64, *d* = 0.6, 95% CI [−17.61, 28.29]) trials for the pain group (congruent, M= −14.23, SD ± 48.50 ms; incongruent, M = 6.22, SD ± 39.46 ms) and controls (congruent, M = 9.66, SD ± 27.09 ms; incongruent, M = 11.56, SD ± 25.32 ms). Table 2 shows the mean response times for the congruent, incongruent and neutral trials for the chronic pain and control groups.

## 4. Discussion 

This study used an eye gaze protocol to consider the attentional biases of individuals with chronic back pain to threatening and neutral images using a modified dot probe paradigm. The eye gaze data highlight important new findings about the differences in attentional bias to threatening and neutral images between individuals with chronic pain and non-symptomatic controls that cannot be identified by using the standard dot probe paradigm. 

Within-group analysis demonstrated that the chronic pain individuals had a significantly higher percentage of fixations, larger pupil diameter and longer average fixation duration on threatening compared to neutral images. There were no significant differences in eye gaze metrics to threatening or neutral images in non-symptomatic controls; the absence of attentional bias suggesting that the back pain specific images were not perceived as threatening for the non-symptomatic controls. Between-group analysis revealed that chronic pain participants also had a longer average fixation duration and larger pupil diameter to threatening images compared to the non-symptomatic controls. Chronic pain patients also had a faster initial fixation to the pain related image compared to the neutral image; the opposite pattern was found for the non-symptomatic controls. In contrast, and of concern to the validity of the standard dot probe procedure, there were no significant differences in attentional biases for the dot probe task between the chronic pain group and non-symptomatic controls. Taken together, the findings of this study suggest that eye gaze metrics may provide a more sensitive measure of attentional bias compared to the dot probe response time. 

The majority of previous research investigating pain-related attentional biases using eye gaze has been within a non-symptomatic population and used word-based stimuli or pain-related faces [13,14,15,16]. To our knowledge, only three studies have previously used back pain-specific images in a dot probe task [25,26,27], but none have used eye-tracking markers concurrent with the dot probe test with pain-related physical activity movements. Consistent with the findings of Roelofs et al. [25] the dot probe data indicated that there was no difference between chronic pain participants and non-symptomatic controls for attention to threatening images in congruent trials. Whereas, the eye gaze behavior in this study demonstrated that chronic pain participants attended to the pain-related images significantly faster, more often and for a significantly longer average duration than neutral images compared to non-symptomatic controls. The attentional bias to threatening stimuli in the pain group may have been due to the implied motion cues within the image. Kourtzi et al. [9] found greater activation of the medial temporal/medial superior temporal cortex (MT/MST) when viewing photographs with implied movement and imaging studies have supported the role of these brain areas in the analysis of movement, but not object, recognition [28]. Action understanding depends, in part, on prior knowledge about the movement’s goal and intention with predictions about an object’s (or body’s) future position being made from the motion implied in the static image [29]. Participant’s memorial biases modulate the increase in attentional bias to threatening information and, therefore, they perceive implied painful motion in the image (e.g., rotation of the back, which causes them pain); the dynamic images accessing a more meaningful motor representation that is presented for analysis through the variety of eye gaze metrics employed in this study.

Eye tracking studies within healthy populations have identified vigilance towards pain-related words and faces [14,15]. In line with previous research [18,19], we identified that chronic pain patients attended to pain related images significantly faster than neutral images and in contrast to the profiles of non-symptomatic controls. Todd et al. [5] have proposed the Threat Interpretation Model of pain that can be used to consider some of the study’s findings. Specifically, there is an initial vigilance towards the threat-related stimuli; as the threat continues, there is an avoidance of the threatening cues. Further, Fashler and Katz [20] identified that both pain and control participants responded faster to neutral images in the dot probe task, whereas the eye movement data demonstrated that in the early phase of the trial, participants attended to the neutral images, then in the later phases, gaze duration was maintained on the injury-related images. These findings are in contrast to those presented here; there were no differences found for reaction time data and chronic pain patients only made faster first fixations to pain related images. Due to the short presentation time used in this study, we were unable to assess whether participants would maintain their gaze on the pain stimuli or, in line with Todd et al., [5] the patients showed an active avoidance of the threatening cues. One possible explanation for the differences could be due to the specificity of the images. Fashler and Katz [20] used images for which both individuals with, and without pain could perceive as threatening (e.g., an open wound, needle inserted into the skin etc.). In contrast, our study used images that were specific to back pain and were not deemed threatening for the control group. Todd et al. [5] propose that initial vigilance occurs when participants interpret the stimuli as threatening. The specificity of the image, and using a homogenous pain group, may, therefore, have influenced the attentional bias to pain related stimuli. Furthermore, the differences between the studies could be due to the image presentation time and type of probe stimulus. In the current study, the images were presented for 500 ms and the dot probe was either.. or : and each probe could appear on either side. In contrast, Fashler and Katz [20] presented the images for 2000 ms and participants were shown a single dot, the response required participants to press one of two buttons to indicate the side of the screen the dot was on. The significant differences in the methods between the studies may explain the differences in the reaction time data. In contrast to Fashler and Katz [20], Yang et al. [30] found that chronic pain participants had an early attentional bias towards catastrophe-based words, followed by avoidance of pain words. Due to the short stimuli presentation time within our study, we were only able to assess initial vigilance, consistent with previous research in pain populations [18]. This bottom up process is considered to be automatic in anxious individuals. In contrast to other studies identifying a vigilance-avoidance pattern [18,30], this study used images of physical activity (rather than words) associated with pain. Images with implied movement provide elements of action understanding. They may give more personal meaning, agency, ownership and motor response [31] for the viewed activity for chronic back pain participants compared to words, pain faces or images of an individual experiencing pain (e.g., someone grimacing while they are completing an activity). Future studies should present pain-related images for a longer duration to identify whether chronic back pain participants maintain vigilance or attend then avoid threatening images. Using images that are specific to the pain condition may also provide researchers with the ability to alter biases to allow for top down processing of attending to goal directed information. This enhanced attention to goal related information may reduce avoidance of activity and disability levels. Although a greater understanding of attention in this context is needed, preliminary evidence has supported the therapeutic benefits of attentional bias modification in pain populations [32,33].

In this study, chronic pain participants showed a significantly larger pupil diameter to threatening images compared to controls. As well as for light intensity, the pupil dilates under conditions of high attentional allocation and also in response to emotionally-congruent information [34,35]. In this regard, it has been suggested that pupil dilation is a physiological response that can indicate brain mechanisms associated with the processing of emotional information [36]. When viewing threatening, emotional images pupil dilation has also been found to be mediated by increased sympathetic activity (e.g., increased heart rate and skin conductance) [34]. According to the biopsychosocial model of pain [37], there is both a psychological and physiological response to pain. If individuals attend to their physiological reaction to emotional information, it increases their worry-based anxiety, based on schemas, causing them to associate the increased arousal with the pain and lead to active avoidance behaviors. The results from this study suggest that not only are individuals with chronic pain attending to pain-related images more than controls, but they are also allocating greater visual attention to them indicating that the stimuli having greater emotional congruence and meaning. This study suggests that investigating pupil diameter could be a useful addition to the study of chronic pain.

The eye gaze data from this study provides support for current models of attention and extends the current chronic pain literature. Models of attentional bias within chronic pain attribute slightly different roles to the process of attention. In general, they propose that individuals in pain are fearful of, and threatened by, pain [2,3,38], which causes them to over attend to pain-related information. Pain is prioritized over other demands for attention [39], which interferes with movement, leads to higher levels of anxiety, catastrophizing and ultimately exacerbation of levels of disability [40]. Understanding the mechanisms of attention to information which is perceived as threatening, is essential to better understand approaches to effective intervention. 

The longer duration of attention to pain-related images in this study may be a function of the schemas associated with the movements. During experimental debriefs, some of the chronic pain participants commented that the images they viewed reminded them of activities in which they would expect to experience pain. Beck [41] proposed that maladaptive schemas cause individuals to have a preoccupation with threatening information and subsequently catastrophize due to a negative interpretive and attentional bias. Pincus and Morley [4] proposed the schema enmeshment model of pain (SEMP) in an attempt to explain recall bias in chronic pain patients through the operation of schemas. Pain schemas contain sensory, intensity, spatial and temporal features of pain, while illness schemas contain information about the consequences of illness, and self-schemas contain information about the self. The chronic pain experience may have illness related schemas about the implications of self-future activity, which causes enmeshment with the pain sensory, self and illness schemas. The anxiety experienced by the pain participants with particular activities they perceive as pain threatening leads to different behavioral and cognitive reactions [1]. The misinterpretation of pain stimuli leads to excessive fear of physical activity and avoidance of physical activity. This may be due to their negative cognitions and increased somatic anxiety about completing the activity, which leads to physical avoidance behavior [42]. Current interventions in the UK tend to focus on reducing a patient’s disability through improved education about their pain and the opportunity to discuss problems with particular activities [43]. Understanding the cognitive and emotional mechanisms behind a patient’s initial activity avoidance behavior may allow for more specific interventions that modify the way chronic pain patients not only attend to activity-related information, but also the way they interpret the planned movement. The interaction of these two cognitive biases will affect the processing of information, lead to behavior change and, potentially, reduce the patient’s disability. 

There were some limitations in the study. First, the PHODA images used were not assessed for their affective content (valence and arousal) or whether the images were personally meaningful to the individual. The images have, however, been used successfully in previous dot probe studies and to assess perceived harmfulness of daily activities [44,45] suggesting they have good ecological validity. Although we did not ask chronic pain participants to rate the images directly, in follow up manipulation checks, pain participants reported that they could attribute the images to behaviors in their daily life and that the activities would be difficult and painful for them to complete at home. Similarly, verbal follow up checks with the control group demonstrated that the PHODA images represented a “neutral” image set and they showed no bias to either image type. Therefore, the biases predicted in the chronic pain group can be attributed to the pain-specific content of the image set. Future studies should assess the valence and arousal of the images in larger and more varied pain populations. Future research should also consider asking patients to rate their personal relevance to them; there may be different attentional biases towards images that are more personally-relevant, compared to those that are not (e.g., see Lang’s meaning propositions within Bioinformational Theory [46]). 

Due to the recruitment strategy, there may be some selection bias. It has been suggested that pain patients who take part in research are often highly motivated, have more severe pain and respond better to treatment [47]. This is a continuing issue for pain research. We were, however, interested in the patients’ cognitive response to threatening stimuli and did not implement an intervention. Future research may wish to consider comparing the health status of patients who volunteer, compared to those who decline to take part in a study. 

Although the power analysis indicated there was sufficient power for this study, the sample size was low in comparison to other studies investigating attentional biases. Therefore, future research should aim to include a larger sample size. It should be noted that although the use of eye gaze is regarded as a more direct assessment of attention, it does not reflect overt attentional engagement. For example, visual attention can occur in the absence of eye movements [48] and eye tracking technology does not measure peripheral vision, which can be used to complete the dot-probe task accurately [17].

Despite these limitations, the present study provides further support for attentional biases towards pain-related images in chronic back pain participants. This finding is supported by additional data from eye gaze measurement techniques that provide a richer and more detailed analysis of the attentional biases in this population. Future studies should be conducted which investigate whether images that involve movement are more reliable at identifying attentional biases in chronic pain patients compared to pain-related words within a dot probe paradigm. Future research should also investigate the relationship between attention and interpretation of pain to provide an updated model to explain the mechanisms associated with patient responses to chronic pain. If the cognitive mechanisms of attention cause individuals with chronic pain to attend and dwell on painful stimuli, interventions that focus on modifying a patient’s attention to goal-related outcomes (e.g., attentional bias modification) may have important beneficial effects on future quality of life.

## Figures and Tables

**Figure 1 jcm-08-00031-f001:**
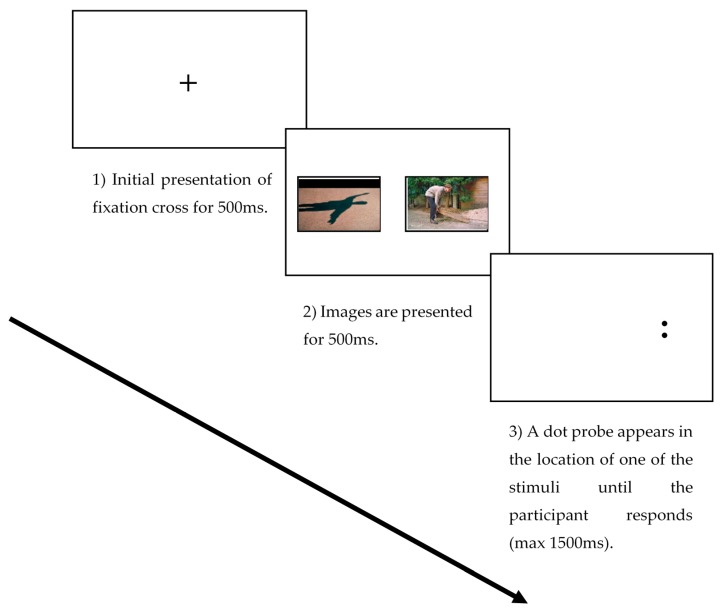
An example of the three stages of the dot probe task in the neutral threat condition.

**Figure 2 jcm-08-00031-f002:**
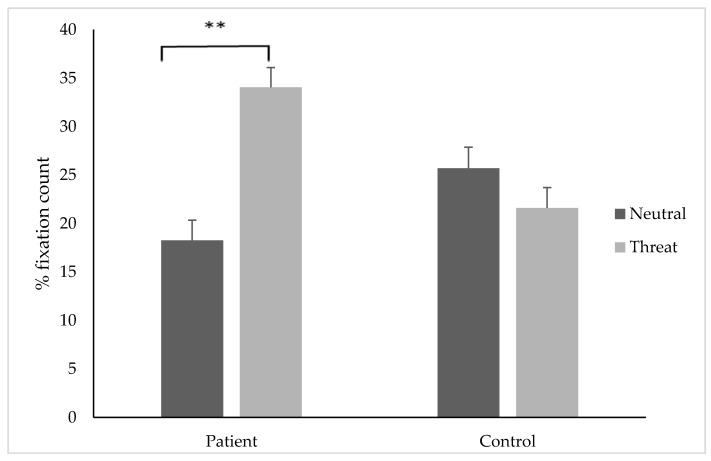
The percentage of fixations (±SE) on threatening or neutral stimuli in the patient and control group. ** indicates *p* < 0.01.

**Figure 3 jcm-08-00031-f003:**
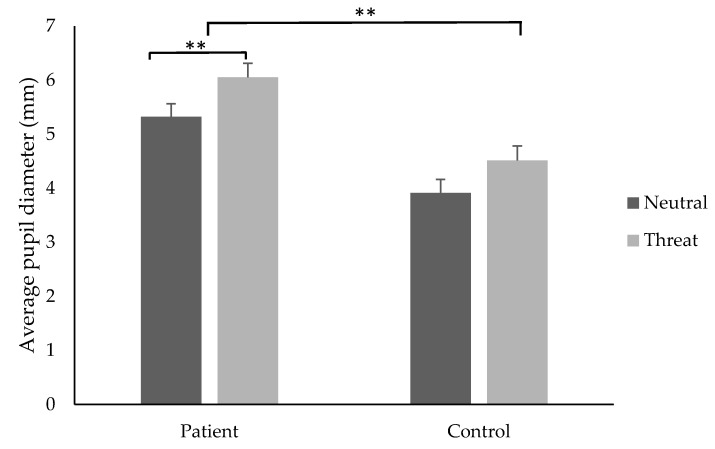
The average pupil diameter (mm) (±SE) for the patient and control groups when attending to threatening and neutral stimuli. ** indicates *p* < 0.01.

**Figure 4 jcm-08-00031-f004:**
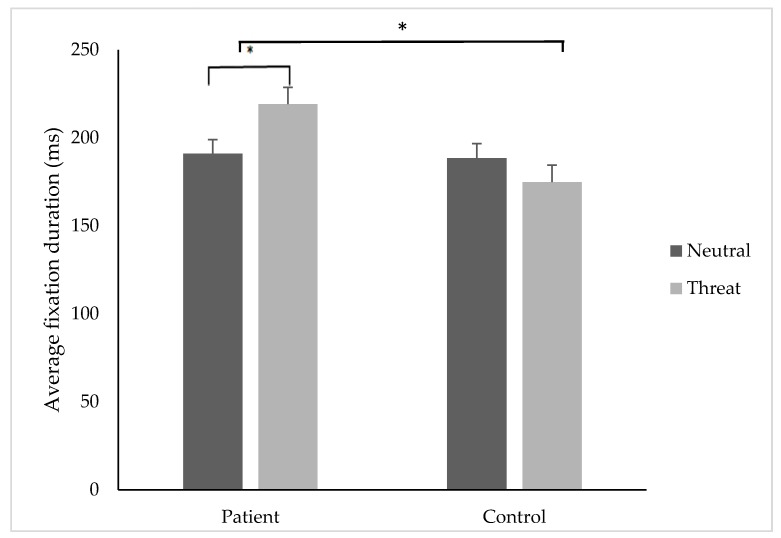
The average fixation (ms) duration (±SE) to threatening and neutral stimuli for the patient and control group. * indicates *p* < 0.05.

**Figure 5 jcm-08-00031-f005:**
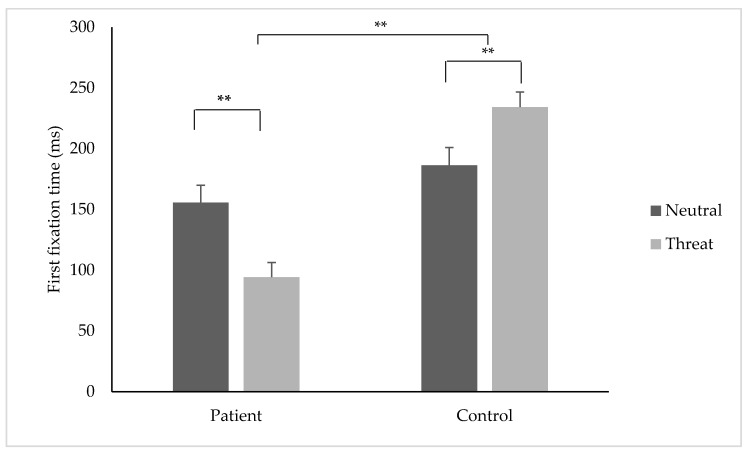
First fixation time (ms) (±SE) to threatening and neutral stimuli for the patient and control group. ** indicates *p* < 0.01.

**Table 1 jcm-08-00031-t001:** Internal consistency measured with Cronbach’s alpha for each outcome variable in the patient and control group.

Group	Outcome Measure	Cronbach’s α
**Patient**	Percentage fixation count	0.750
	Pupil diameter	0.802
	Average fixation duration	0.644
	First fixation time	0.722
**Control**	Percentage fixation count	0.927
	Pupil diameter	0.743
	Average fixation duration	0.616
	First fixation time	0.725

**Table 2 jcm-08-00031-t002:** Mean reaction times of congruent and incongruent trials (in ms; standard deviations in brackets) for the threat and neutral images in the dot probe task for participants with chronic low back pain and the non-symptomatic control group.

	Stimuli	Chronic Pain Group	Non-Symptomatic Control Group
Congruent	Threat (ms)	595.79 (59.40)	548.74 (49.08)
	Neutral (ms)	591.32 (65.10)	540.44 (49.65)
Incongruent	Threat (ms)	597.31 (59.64)	552.69 (48.36)
	Neutral (ms)	591.32 (65.10)	540.44 (49.65)

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
