# Peer review of "Eye Gaze Markers Indicate Visual Attention to Threatening Images in Individuals with Chronic Back Pain"

_jcm, 2018, doi:10.3390/jcm8010031_

Reviewer 1 Report

 There is very little research using gaze behaviour in pain, but this study would be of more interest, if it were better situated in the appropriate literature and dealt with some of limitations more thoroughly and transparently. 

1. The authors cite work by Eysenck, which is highly influential theoretical work in the area of anxiety. However, there are numerous models of chronic pain which are more specific and make similar predictions, such as recent updating of the fear avoidance model (e.g. Crombez G, Eccleston C, Van Damme S, Vlaeyen JW, Karoly P. Fear-avoidance model of chronic pain: the next generation. Clin J Pain 2012;28:475–83.). There is no need to rely on anxiety models when appropriate and more specific theoretical models are available.

2. The authors are over-stating the importance of pictures. There is a more recent meta-analysis (Todd et al., Health Psychology Review, 2018) that finds that both pain-related words and pictures give rise to attentional biases, and the authors should update this section of their literature review, but there is no evidence that the effect sizes are larger for pictures than words. Having said that, the use of pictures with implied movement would be an important research question, but it isn’t one asked by this team. I would focus on the actual contribution here which is the use of gaze behaviour with pain-related pictures.

3. The gaze behaviours that were utilised were a little unusual, so the authors used percentage of fixations, but typically authors use percentage of first fixations? This is because proportion of (or latency to) first fixation is an indicator of potential vigilance (i.e. the allocation of attention). I think that the authors would benefit from engaging with the eye tracking literature more strongly and should read the following: (Cisler, J. M., & Koster, E. H. (2010). Mechanisms of attentional biases towards threat in anxiety disorders: An integrative review. Clinical psychology review30(2), 203-216.). This manuscript very clearly outlines the mechanisms of attention. Can the authors justify why the number of fixations (or percentage) is a clearer measure than the first fixation of attentional processes?

4. The authors do not provide a justification for the use of 500 msec as a presentation time. Research shows that the reliability of gaze behaviour increases as the presentation time increases. (see Skinner, I. W., Hübscher, M., Moseley, G. L., Lee, H., Wand, B. M., Traeger, A. C., ... & McAuley, J. H. (2018). The reliability of eyetracking to assess attentional bias to threatening words in healthy individuals. Behavior research methods50(5), 1778-1792.). These authors recommend presentation times >2000 mseconds (and preferably higher). Can the authors (a) justify the use of such a short presentation time; and (b) provide data on the reliability of their task.

5. The authors have conducted a post-hoc power analysis, which is not really appropriate. Moreover, they report the beta, which is typically achieved power in this context, but does not relate to their claims of power. These data are somewhat dubious. It is a very small sample size and should be claimed as a limitation.

6. The mean percentage of fixations is very low (in both conditions indicating, if I understand the data correctly) that there fixations made on less than 50% of target trials. This likely relates to point 4 above. Could the authors comment?

7. The authors state that there is no significant effect for reaction times, but with such a small sample size, a p = 0.08 suggests that there is likely to be quite a large effect, even though it is not significant. The authors should calculate all ESes plus 95% confidence intervals throughout the manuscript.

8. The same comments that were made about the introduction and results are needed to be changed here (i.e. rely less on the anxiety literature, and discuss limitations more). a

Author Response

We would like to thank the reviewer for their detailed and constructive comments. We have attempted to address all of your comments and feel they have enhanced the paper. If you require further detail or clarification, please let us know.

There is very little research using gaze behaviour in pain, but this study would be of more interest, if it were better situated in the appropriate literature and dealt with some of limitations more thoroughly and transparently. 

The authors cite work by Eysenck, which is highly influential theoretical work in the area of anxiety. However, there are numerous models of chronic pain which are more specific and make similar predictions, such as recent updating of the fear avoidance model (e.g. Crombez G, Eccleston C, Van Damme S, Vlaeyen JW, Karoly P. Fear-avoidance model of chronic pain: the next generation. Clin J Pain 2012;28:475–83.). There is no need to rely on anxiety models when appropriate and more specific theoretical models are available.

We appreciate the reviewer’s point of view and had used Eysenck’s model as pain patients tend to experience high levels of anxiety when they are faced with physical activity. However, we have decided to remove references to this theory, and used other models of chronic pain instead. This has resulted to changes in the introduction and discussion.

The authors are over-stating the importance of pictures. There is a more recent meta-analysis (Todd et al., Health Psychology Review, 2018) that finds that both pain-related words and pictures give rise to attentional biases, and the authors should update this section of their literature review, but there is no evidence that the effect sizes are larger for pictures than words. Having said that, the use of pictures with implied movement would be an important research question, but it isn’t one asked by this team. I would focus on the actual contribution here which is the use of gaze behaviour with pain-related pictures.

Unfortunately this manuscript hadn’t been published when we submitted this study, and we would like to thank the reviewer for recommending it to us. We have amended the introduction in line with your comments to focus on the use of gaze behaviour and pain related pictures Page 2, lines 46-49.

The gaze behaviours that were utilised were a little unusual, so the authors used percentage of fixations, but typically authors use percentage of first fixations? This is because proportion of (or latency to) first fixation is an indicator of potential vigilance (i.e. the allocation of attention). I think that the authors would benefit from engaging with the eye tracking literature more strongly and should read the following: (Cisler, J. M., & Koster, E. H. (2010). Mechanisms of attentional biases towards threat in anxiety disorders: An integrative review. Clinical psychology review30(2), 203-216.). This manuscript very clearly outlines the mechanisms of attention. Can the authors justify why the number of fixations (or percentage) is a clearer measure than the first fixation of attentional processes?

We thank the reviewer for directing our attention to this article and although this article is focused on the attentional mechanisms of anxiety disorders, we agree that it does give an indication for the metrics we should be using. We received a similar comment from one of the other reviewers and have decided to also include first fixation time. This data has been added into the results section and integrated into the discussion. We hope this addresses your comment.

The authors do not provide a justification for the use of 500 msec as a presentation time. Research shows that the reliability of gaze behaviour increases as the presentation time increases. (see Skinner, I. W., Hübscher, M., Moseley, G. L., Lee, H., Wand, B. M., Traeger, A. C., ... & McAuley, J. H. (2018). The reliability of eyetracking to assess attentional bias to threatening words in healthy individuals. Behavior research methods50(5), 1778-1792.). These authors recommend presentation times >2000 mseconds (and preferably higher). Can the authors (a) justify the use of such a short presentation time; and (b) provide data on the reliability of their task.

We thank the reviewer for directing our attention to this article. The shorter presentation time was used to assess attentional initial vigilance. We have included in the discussion that we were only able to assess initial vigilance due to the short presentation time and that a longer duration is needed to assess avoidance or maintenance of gaze. Based on this article, we would look to include a longer presentation time in future studies. We have included data on the internal consistency of the eye tracking data (Page 6, lines 220-225).

The authors have conducted a post-hoc power analysis, which is not really appropriate. Moreover, they report the beta, which is typically achieved power in this context, but does not relate to their claims of power. These data are somewhat dubious. It is a very small sample size and should be claimed as a limitation.

We have included this as a limitation in the discussion (Page 11, Line, 368-370)

The mean percentage of fixations is very low (in both conditions indicating, if I understand the data correctly) that there fixations made on less than 50% of target trials. This likely relates to point 4 above. Could the authors comment?

The reviewer is correct in their analysis, and we agree that this could be a consequence of the short presentation time. This finding may be similar to other studies which look at longer durations. In order to account for this possibility we included 100 trials, as previous studies have found this as a limitation of other dot probe studies. In future we will include longer presentation times.

The authors state that there is no significant effect for reaction times, but with such a small sample size, a p = 0.08 suggests that there is likely to be quite a large effect, even though it is not significant. The authors should calculate all ESes plus 95% confidence intervals throughout the manuscript.

Although we agree that the sample size is low, the dot probe paradigm has been demonstrated to not be a reliable measure of attention due to the reaction time element. We have gone through the results section and made sure that the effect sizes and confidence intervals are included throughout all of the manuscript.

The same comments that were made about the introduction and results are needed to be changed here (i.e. rely less on the anxiety literature, and discuss limitations more).

We thank the reviewer for these comments and have made the changes suggested throughout both the introduction and the discussion.

Reviewer 2 Report

The current study sought to examine attention to threatening images using dotprobe methodology and eye-tracking amongst a sample of chronic back pain patients and a matched healthy sample. Dotprobe analyses revealed no significant differences between samples. Eye tracking analyses indicated that chronic pain patients demonstrated more fixations, larger pupil diameter and longer gaze duration towards threatening images compared to neutral image. Furthermore, gaze duration pupil diameter towards threatening images was also higher amongst the chronic pain patients compared to the healthy sample.

While the current findings attest to the utility of employing eye tracking methods rather than manual response tasks that provide a snap shot of attention, I believe there are some issues that need to be addressed to further improve the current paper.

The authors accurately describe the necessity of using ecologically valid stimuli (movement related stimuli). within attention bias research. It should also be highlighted  within the introduction that these should also be ‘personally relevant’. Further, it is unclear whether pictures were indeed personally relevant for the chronic back pain patients. I believe  this was not assessed. This should be noted more explicitly as limitation. Also, thoughts about how this could be examined or how stimuli could be made even more personally salient should be added to the discussion.

Page 2: word missing in sentence beginning at line 46.

It would be helpful for the reader  to describe the findings referred to at page 2 (reference 21,22; line 78). What did these eye tracking findings reveal? And how do these relate  to the current findings. The latter could be elaborated upon in the discussion.

It is not entirely clear why the authors expect that chronic pain patients would show more fixations to threatening images compared to controls aside of the hypothesis that they would also show longer gaze duration. Number of fixations and gaze duration are likely highly correlated? What is the rationale to examine both number of fixations and overall gaze duration: this should be spelled out more clearly. Also, it is unclear why there were not measures of initial fixation (early attention to pain) such as probability of first fixation as is  usually done in other eye  tracking studies examining attention to pain. I suggest to calculate this and add this to the results section.

The sample size is very low. Also, no information is provided on response rate and reasons to refuse participation. How many were invited, how many declined, and for what reasons? This should be added.

Materials: I suggest to first describe the dot-probe task and then the eye  tracking task. At present it appears as if these are two different task whereas it is actually a dot-probe task during which eye movements were also assessed. Changing the order of description and referring to eye tracking being implemented within the dotprobe assessment would make this more clear.

The authors conclude that only initial vigilance was assessed. However, the stimulus presentation time was set at 500 ms. Previous dot-probe paradigms have used shorter presentation times: this should be acknowledged. Further, the measure of initial vigilance is currently missing: probability of first fixation (see also comment 4).

The authors conclude that the current study provides support for current models of attention and that it extends the chronic pain literature. They discuss relationships with interpretation biases and memory biases; however, it is unclear in what way the current findings relate to both type of biases as these were not assessed in the current study. I suggest to rephrase this paragraph in terms of what could be future avenues for research.

P 10, line 336: typo: logical validity should be ‘ecological’ validity

P 10 line 336, ‘have’ is  erroneously mentioned twice.

Author Response

We would like to thank the reviewer for their detailed and constructive comments. We have attempted to address all of your comments and feel they have enhanced the paper. If you require further detail or clarification, please let us know.

The current study sought to examine attention to threatening images using dotprobe methodology and eye-tracking amongst a sample of chronic back pain patients and a matched healthy sample. Dotprobe analyses revealed no significant differences between samples. Eye tracking analyses indicated that chronic pain patients demonstrated more fixations, larger pupil diameter and longer gaze duration towards threatening images compared to neutral image. Furthermore, gaze duration pupil diameter towards threatening images was also higher amongst the chronic pain patients compared to the healthy sample.

While the current findings attest to the utility of employing eye tracking methods rather than manual response tasks that provide a snap shot of attention, I believe there are some issues that need to be addressed to further improve the current paper.

The authors accurately describe the necessity of using ecologically valid stimuli (movement related stimuli). within attention bias research. It should also be highlighted  within the introduction that these should also be ‘personally relevant’. Further, it is unclear whether pictures were indeed personally relevant for the chronic back pain patients. I believe  this was not assessed. This should be noted more explicitly as limitation. Also, thoughts about how this could be examined or how stimuli could be made even more personally salient should be added to the discussion.

We agree with the reviewer that the images should be personally relevant to the individual. Unfortunately we didn’t ask the patients to rate the images for their personal relevance. We had already mentioned this as a limitation, however we have made this more explicit in the limitations section of the discussion (page 10, line 347-348, 357-359).

Page 2: word missing in sentence beginning at line 46.

‘Have demonstrated’ has been added in.

It would be helpful for the reader  to describe the findings referred to at page 2 (reference 21,22; line 78). What did these eye tracking findings reveal? And how do these relate  to the current findings. The latter could be elaborated upon in the discussion.

We have expanded upon these two studies, and also added in another one that was recommended by one of the other reviewers (Page 3, 93-113)

It is not entirely clear why the authors expect that chronic pain patients would show more fixations to threatening images compared to controls aside of the hypothesis that they would also show longer gaze duration. Number of fixations and gaze duration are likely highly correlated? What is the rationale to examine both number of fixations and overall gaze duration: this should be spelled out more clearly. Also, it is unclear why there were not measures of initial fixation (early attention to pain) such as probability of first fixation as is  usually done in other eye  tracking studies examining attention to pain. I suggest to calculate this and add this to the results section.

We have attempted to make it clearer in the data analysis section why we are looking at number of fixations and average fixation duration. We can see that how it was written previously makes it sound as though the overall gaze duration was investigated, however, average fixation duration is a measure of the average duration of an individual fixation. The purpose of including this measure in addition to % of fixations is to identify whether participants not only made more fixations on a particular image type, but that they also held their gaze there for a longer

The sample size is very low. Also, no information is provided on response rate and reasons to refuse participation. How many were invited, how many declined, and for what reasons? This should be added.

We have included the low sample size as a limitation in the discussion. Unfortunately we don’t have the exact data on the number of individuals invited, declined and for what reason. This is certainly a variable we would look to include in future research studies.

Materials: I suggest to first describe the dot-probe task and then the eye  tracking task. At present it appears as if these are two different task whereas it is actually a dot-probe task during which eye movements were also assessed. Changing the order of description and referring to eye tracking being implemented within the dotprobe assessment would make this more clear.

We thank the reviewer for this comment and have made the changes requested which I hope will make it clearer for the reader.

The authors conclude that only initial vigilance was assessed. However, the stimulus presentation time was set at 500 ms. Previous dot-probe paradigms have used shorter presentation times: this should be acknowledged. Further, the measure of initial vigilance is currently missing: probability of first fixation (see also comment 4).

In line with the comment above we have included the first fixation time in the results section.

The authors conclude that the current study provides support for current models of attention and that it extends the chronic pain literature. They discuss relationships with interpretation biases and memory biases; however, it is unclear in what way the current findings relate to both type of biases as these were not assessed in the current study. I suggest to rephrase this paragraph in terms of what could be future avenues for research.

I have attempted to alter this paragraph to ensure that the conclusion is focused on the attentional bias

P 10, line 336: typo: logical validity should be ‘ecological’ validity

This has been changed.

P 10 line 336, ‘have’ is  erroneously mentioned twice.

We removed one of the have’s.

Reviewer 3 Report

This is a potentially interesting study investigating attentional biases in people with chronic pain. To the extent that most previous studies on this matter tend to be based on reaction times (RT) using dot-probe paradigms with word or face stimuli, the use of eye-tracking methods (allowing a more direct method of assessment of attentional processes) and images of back pain stimuli in this dot probe experiment is an important development and adds to research in this area. My enthusiasm for the study is dampened, however, by the complete absence of information about pain-related/psychosocial function in the pain sample – patients with chronic pain are often heterogeneous with respect to psychosocial function and factors such as fear of pain can modulate attentional bias to threatening stimuli in people with and without chronic pain. There are some other problems concerning the methodology/analysis and interpretation which need to be addressed.

Major points

1. Introduction. The Introduction develops the rationale for investigating attentional biases via eye movements in patients with chronic back pain for neutral and pain-related pictorial stimuli. I was surprised, however, that there was no mention here, or in the Discussion, of the eye movement study of Fashler et al. (2016) where responses from individuals with and without (predominantly bodily) chronic pain were compared on a dot-probe task using injury-related and neutral pictures.

2. Methods, Participants (pp. 2-3). Is there any data related to pain-related and/or psychosocial function within the CP patient group? CP patients are often heterogeneous in their psychosocial presentation and information concerning patients’ level of pain disability, health anxiety, pain catastrophizing, hypervigilance and/or fear of pain would be enormously helpful, especially considering their potential modulatory role in (selective) attentional processing of threatening environmental stimuli in people with chronic pain (e.g., Asmundson & Hadjistavropoulos, 2007; Yang et al., 2013). If not, at the very least, this needs to be recognised as a limitation.

3. Methods  p.5 Lines 180-186. The post-hoc power analyses are difficult to follow. Do the calculations for each of the dependent variables refer to main effects of group or group x stimuli type interaction effects – the latter would be more informative (in any case, the stated value for the dot probe data suggests it may be underpowered to detect effects)? The sample size for the study is relatively small and only large (interaction) effects are likely to be observed.

4. Results. Were there any data concerning the reliability (e.g., internal consistency) of eye movement outcome measures?  This may be especially relevant given the stimuli exposure time was relatively brief (longer presentation times increase reliability estimates; Skinner et al., 2017). 

5. Discussion. There is little in the Discussion as to why patterns of eye movement for threatening stimuli were different in patients with CP compared to controls but reaction time on the dot probe did not discriminate between the groups. While the authors briefly make the point that this could be indicative of the greater sensitivity of eye gaze metrics than dot probe reaction time (p.8), this needs elaboration – also, other studies using dot probe in CP populations have observed differences in attentional bias as measured by reaction time (Schoth et al., 2012). Some explanation for these discrepancies is warranted (e.g., stimulus presentation time).

6. Discussion p.10 Lines 336-341. The image manipulation checks are important – but need to be appropriately detailed and placed in the Results section rather than introduced in the Discussion.

Minor comments:

1. Introduction p.2 Lines 44-45. The statement beginning ‘Anxiety is reported as consistently as both a comorbid condition associated with anxiety…’ does not make sense. Please rephrase.

2. Introduction p.2 Lines 46-47. The statement beginning ‘Meta-analyses that pain participants…’ is missing a word.

3. Introduction p.2 Line 69. Ned to remove an ‘also’ from the statement ‘In addition, eye tracking also has the ability to also…’ beginning

4. Methods p.5 Data analysis.  Please spell out ANOVA (analysis of variance) and state the adopted alpha value for the study (e.g., p < .05).

5. Results pp.6-7 Figures 2, 3, 4. The graphs are clear but would be easier to interpret if error bars represented standard errors or 95% confidence intervals (rather than SDs).

6. Results p.6 Lines 203, 208. p values should be<0.001 rather than ‘0.000’ or ‘<=0.00’.< p="">

7. Results p.7 Lines 214-215. The p value for the significant interaction is stated as =0.05. Is the adopted alpha level<0.05 or another value? If the former, this needs clarification (e.g., if p value has been rounded up then maybe use 3 decimal places for p values throughout?). Also, the significant main effect of stimuli type on fixation duration is surprising given the longer duration for threat stimuli in patients but (numerically) shorter duration for threat stimuli in controls. 

8. Discussion pp.9-10 Lines 308-331. Some mention of the Schema Enmeshment Model of Pain (SEMP; Pincus & Morley, 2001) is appropriate given the focus here on cognitive schemata and their role in information processing biases in chronic pain.

9. Discussion p.10 Lines 333-349. Although the use of eye-tracking is often regarded as a more direct assessment of attention, it does not exactly reflect overt attentional engagement. For example, visual attention can occur in the absence of eye movements (Zhao et al., 2012) and eye movements typically do not measure peripheral vision, which can be used to complete the dot-probe task (Vervoort et al., 2013). This point should be acknowledged.

References

Asmundson GJ, Hadjistavropoulos HD. Is high fear of pain associated with attentional biases for pain-related or general threat? A categorical reanalysis. The Journal of Pain. 2007;8(1):11-18.

Fashler SR, Katz J. Keeping an eye on pain: investigating visual attention biases in individuals with chronic pain using eye-tracking methodology. Journal of Pain Research. 2016;9:551-561.

Pincus T, Morley S. Cognitive-processing bias in chronic pain: a review and integration. Psychological Bulletin. 2001; 127(5):599-617.

Schoth DE, Nunes VD, Liossi C. Attentional bias towards pain-related information in chronic pain; a meta-analysis of visual-probe investigations. Clinical Psychology Review. 2012;32(1):13-25.

Skinner IW, Hübscher M, Moseley GL, Lee H, Wand BM, Traeger AC, et al. The reliability of

eyetracking to assess attentional bias to threatening words in healthy individuals. Behav Res

Methods. 2017;50(5):1778–92.

Vervoort T, Trost Z, Prkachin KM, Mueller SC. Attentional processing of other’s facial display of pain: an eye tracking study. Pain. 2013;154(6):836–44.

Yang Z, Jackson T, Chen H. Effects of chronic pain and pain-related fear on orienting and maintenance of attention: an eye movement study. The Journal of Pain. 2013;14(10):1148-1157. 

Zhao M, Gersch TM, Schnitzer BS, Dosher BA, Kowler E. Eye movements and attention: the role of pre-saccadic shifts of attention in perception, memory and the control of saccades. Vision Research. 2012;74: 40–60.

Author Response

We would like to thank the reviewer for their detailed and constructive comments. We have attempted to address all of your comments and feel they have enhanced the paper. If you require further detail or clarification, please let us know.

This is a potentially interesting study investigating attentional biases in people with chronic pain. To the extent that most previous studies on this matter tend to be based on reaction times (RT) using dot-probe paradigms with word or face stimuli, the use of eye-tracking methods (allowing a more direct method of assessment of attentional processes) and images of back pain stimuli in this dot probe experiment is an important development and adds to research in this area. My enthusiasm for the study is dampened, however, by the complete absence of information about pain-related/psychosocial function in the pain sample – patients with chronic pain are often heterogeneous with respect to psychosocial function and factors such as fear of pain can modulate attentional bias to threatening stimuli in people with and without chronic pain. There are some other problems concerning the methodology/analysis and interpretation which need to be addressed.

  Major points

Introduction. The Introduction develops the rationale for investigating attentional biases via eye movements in patients with chronic back pain for neutral and pain-related pictorial stimuli. I was surprised, however, that there was no mention here, or in the Discussion, of the eye movement study of Fashler et al. (2016) where responses from individuals with and without (predominantly bodily) chronic pain were compared on a dot-probe task using injury-related and neutral pictures.

We thank the reviewer for directing our attention towards this article and we have included it in both the introduction and the discussion. Page 2, lines 93- 102 and Page 11, lines 344-362

2. Methods, Participants (pp. 2-3). Is there any data related to pain-related and/or psychosocial function within the CP patient group? CP patients are often heterogeneous in their psychosocial presentation and information concerning patients’ level of pain disability, health anxiety, pain catastrophizing, hypervigilance and/or fear of pain would be enormously helpful, especially considering their potential modulatory role in (selective) attentional processing of threatening environmental stimuli in people with chronic pain (e.g., Asmundson & Hadjistavropoulos, 2007; Yang et al., 2013). If not, at the very least, this needs to be recognised as a limitation.

We had collected data relating to the individuals fear of movement, however, we had originally decided not to include it because due to the low sample size, we would not have been able to do any analysis to investigate whether fear of pain modulated attentional biases. The data shows that the chronic pain group had significantly higher fear of movement compared to the control group. We can add this data in if the reviewer would like us too, please advise on what you would like us to do.

 3. Methods  p.5 Lines 180-186. The post-hoc power analyses are difficult to follow. Do the calculations for each of the dependent variables refer to main effects of group or group x stimuli type interaction effects – the latter would be more informative (in any case, the stated value for the dot probe data suggests it may be underpowered to detect effects)? The sample size for the study is relatively small and only large (interaction) effects are likely to be observed.

A similar comment was made by another reviewer, therefore we have made it clearer in the discussion that the sample is small

 4.  Results. Were there any data concerning the reliability (e.g., internal consistency) of eye movement outcome measures?  This may be especially relevant given the stimuli exposure time was relatively brief (longer presentation times increase reliability estimates; Skinner et al., 2017).  

We have now included reliability data for the eye movement outcome measures for each group (Page 6, lines 220-225).

 5. Discussion. There is little in the Discussion as to why patterns of eye movement for threatening stimuli were different in patients with CP compared to controls but reaction time on the dot probe did not discriminate between the groups. While the authors briefly make the point that this could be indicative of the greater sensitivity of eye gaze metrics than dot probe reaction time (p.8), this needs elaboration – also, other studies using dot probe in CP populations have observed differences in attentional bias as measured by reaction time (Schoth et al., 2012). Some explanation for these discrepancies is warranted (e.g., stimulus presentation time). 

We have tried to elaborate on this in more detail to explain some of these discrepancies. Page 11, lines 344-362.

 6. Discussion p.10 Lines 336-341. The image manipulation checks are important – but need to be appropriately detailed and placed in the Results section rather than introduced in the Discussion.  

We can see why the reviewer would suggest this, however, this process was not a formal image manipulation check, but was a series of informal comments made by the participants. We accept that not including any image manipulation checks is a limitation of this study and is something we would ensure is included in future research.

Minor comments:

Introduction p.2 Lines 44-45. The statement beginning ‘Anxiety is reported as consistently as both a comorbid condition associated with anxiety…’ does not make sense. Please rephrase.

We have rephrased this sentence to make it clear for the reader.

2. Introduction p.2 Lines 46-47. The statement beginning ‘Meta-analyses that pain participants…’ is missing a word.

We have included ‘have demonstrated’ to make it clear for the reader.

3. Introduction p.2 Line 69. Ned to remove an ‘also’ from the statement ‘In addition, eye tracking also has the ability to also…’ beginning

The word also was removed.

4. Methods p.5 Data analysis.  Please spell out ANOVA (analysis of variance) and state the adopted alpha value for the study (e.g., p < .05).

ANOVA was spelled out and the alpha value included.

5. Results pp.6-7 Figures 2, 3, 4. The graphs are clear but would be easier to interpret if error bars represented standard errors or 95% confidence intervals (rather than SDs).

All graphs have been changed to include the standard error instead of the SD. The SD is still reported within the text.

6. Results p.6 Lines 203, 208. p values should be<0.001 rather than ‘0.000’ or ‘<=0.00’.< span="">

p values are reported as p< 0.0001

7. Results p.7 Lines 214-215. The p value for the significant interaction is stated as =0.05. Is the adopted alpha level<0.05 or another value? If the former, this needs clarification (e.g., if p value has been rounded up then maybe use 3 decimal places for p values throughout?). Also, the significant main effect of stimuli type on fixation duration is surprising given the longer duration for threat stimuli in patients but (numerically) shorter duration for threat stimuli in controls. 

The statistics have been checked over and all significant values have been made clear. In line with other research studies, the alpha level adopted is p≤ 0.05.

8. Discussion pp.9-10 Lines 308-331. Some mention of the Schema Enmeshment Model of Pain (SEMP; Pincus & Morley, 2001) is appropriate given the focus here on cognitive schemata and their role in information processing biases in chronic pain.

This model of pain has been included in the discussion.

9. Discussion p.10 Lines 333-349. Although the use of eye-tracking is often regarded as a more direct assessment of attention, it does not exactly reflect overt attentional engagement. For example, visual attention can occur in the absence of eye movements (Zhao et al., 2012) and eye movements typically do not measure peripheral vision, which can be used to complete the dot-probe task (Vervoort et al., 2013). This point should be acknowledged.

We have acknowledged this point in the limitations section.

Round  2

Reviewer 1 Report

The manuscript has been improved significantly and I believe is now ready to be published.

Reviewer 2 Report

The authors have adequately revised the manuscript.